# A Hierarchical Hyper-rectangle Mass Model for Fine-grained Entity Typing

## Abstract

Fine-grained entity typing is the task of detecting types of entities inside a given language text. Entity typing models typically transform entities into vectors in high-dimensional space, hyperbolic space, or add additional context information. However, such spaces or feature transformations are not compatible with modeling types' inter-dependencies and diverse scenarios. We study the ability of the hierarchical hyper-rectangle mass model(hRMM), which represents mentions and types into hyper-rectangles mass(hRM) and thus captures the relationships of ontology into a geometric mass view. Natural language contexts are fed into encoder and then projected to hyper-rectangle mass embedding(hRME). We find that hRM perfectly depicts features of mentions and types. With further research in hypervolume indicator and adaptive thresholds, performance achieves additional improvement. Experiments show that our approach achieves better performance on several entity typing benchmarks and attains state-of-the-art results on two benchmark datasets.

## 1 Introduction

Entity typing is the task of assigning types to named entities in language texts. Entity typing has shown to be widely used in tasks such as entity linking (Dai et al., 2019), knowledge base learning (Hao et al., 2019), and sentence classification. In recent years, this task has become a major focus of NLP research.

Many classification systems or hierarchical models have been proposed and achieved promising results in entity typing tasks. Geometric embedding and multi-classification learning are two main techniques in recent years. Rather than representing objects with vectors, geometric representation models are recently assumed to be more suited to expressing relationships in the domain.

Geometric embedding such as Box embedding (Onoe et al., 2021) uses box space to represent mention and types (Vilnis et al., 2018). Box embedding is an interesting view which has been employed for knowledge graph reasoning (Ren et al., 2020), knowledge graph completion (Abboud et al., 2020), and joint hierarchical representation (Patel et al., 2020). However, it has a few drawbacks: (1) representation of box embedding is too simple to capture latent features in mention and types (2) box embedding treats entity typing task as a multiclass or multi-label classification task, which means it cannot learn hierarchical knowledge.

Apart from box embedding, Learning hierarchical knowledge in euclidean space (Chen et al., 2020), (Yogatama et al., 2015) or hyperbolic space (López & Strube, 2020) are not perfect and far from enough for representing entities which dive into extremely diverse scenarios. In addition, these methods are not capable to illustrate hierarchical relationships between mention, supertype, and subtype.

To overcome the aforementioned drawbacks, we build an entity typing model named hRMM for the entity typing task. As illustrated in figure2, our model builds an hRM hierarchical architecture that represents entity types and mentions into hRME (Figure 1).

Compared with the baseline model – box embedding (Onoe et al., 2021), we add density and size scale parameters to determine the mass and size of types in geometric view. These parameters can further investigate more latent features in the language context.

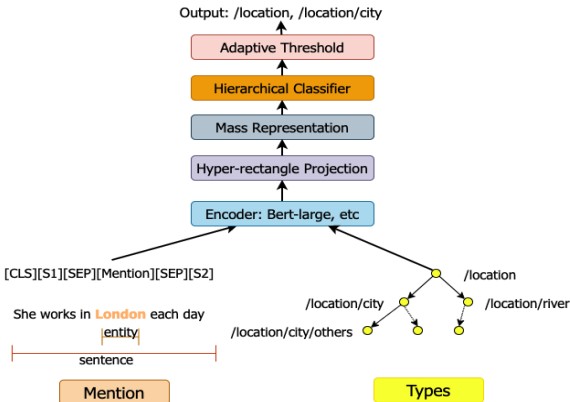

Figure 1: hRMM architecture for predicting types of entity mention "London" in a given sentence

Further, we develop hypervolume indicators and adaptive thresholds which bring additional improvements. The hypervolume indicator minimizes the combination of losses to achieve better loss results. Adaptive threshold, a greedy threshold method, also outperformed the normal threshold method.

Without any hand-crafted features or data processing methods, experiments on benchmark datasets including Figer, UFET, and OntoNotes demonstrate that our approach outperforms baseline models and prior works. It also indicates that our approach is capable of capturing the latent hierarchical structure and language feature in entity typing tasks. We will publish all source codes and datasets of this work on GitHub for further exploration.

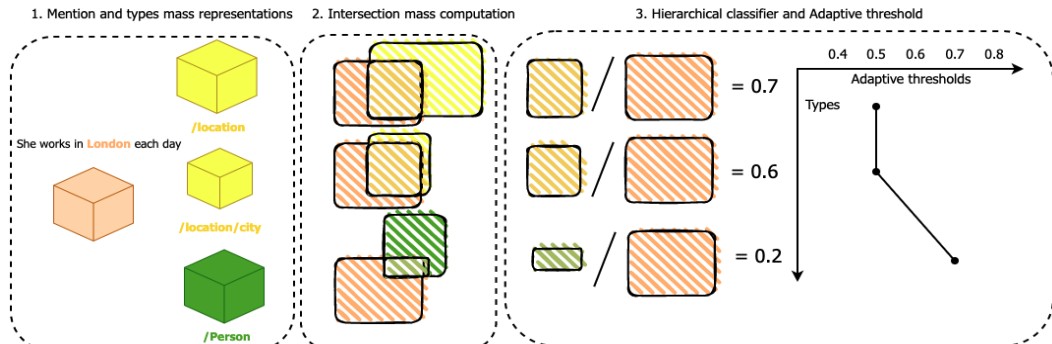

Figure 2: Illustration of hRM generation and intersection computations. (Colors for objects represent density value which is linear to RGB color model(Ibraheem et al., 2012))

## 2    RELATED WORKS

Many efforts have been invested in entity typing tasks for years. In addition to the related work discussed in the previous section, few works focus on correcting noisy labels to improve metrics. Due to the establishment of large-scale weakly and distantly supervised annotation data, it may contain abundant noise and thus severely hinder the performance of the entity typing task.

Co-teaching(Han et al., 2018) simultaneously train two collaborating networks to filter out potentially noisy labels according to their loss. DivideMix (Li et al., 2020) leveraged a one-dimensional and two-component Gaussian Mixture Model (GMM) to model the loss distribution of clean and noisy labels.SELF (Nguyen et al., 2019) selects the clean labels according to the agreement between annotated labels and the network's prediction. However, we found that the loss does not form a bimodal distribution in entity typing tasks and thus it is hard to distinguish clean and noisy labels

precisely. In addition, for deep learning model, precision is not that higher than restricted rules to find noisy labels.

Our work differs from these works in that we do not rely on a deep learning model to denoise the distantly-labeled data. Parameters in our model such as density and size scale are trained to depict hierarchies of types. These parameters restrict the hRM and thus denoise the noisy data (show in 3.3). Compared with the deep learning model, our restricted approach holds higher precision. Further, according to the experiment in this paper, hRMM learns hierarchical knowledge relies on prior auxiliary resources and the result outperforms prior works in the loss task. In summary, our work focuses on improving the representation of hierarchical entities for hRMM. To this end, we integrate our model in three novel ways: mass representation of mention span and types, hierarchical learning loss, and adaptive threshold.

## 3 MODEL

In this section, we introduce hRMM, our proposed model for entity typing tasks. We start with the entity typing task definition, followed by the overview of hRMM (shown in Figure 1). Further, we introduce technical details of hRME representations includes mention representation, and hierarchical type representation. Apart from descriptions of representation, hierarchical classifier, hierarchical learning loss, and adaptive thresholds will also be discussed at the end of this section.

### 3.1 PROBLEM DEFINITION

Entity typing is considered a hierarchical multi-label classification task. Entity typing datasets consist of a collection of sentence text $\mathcal{S}$ with mention span $\mathcal{X}$, entity $\mathcal{E}$, and a set of hierarchical types $\mathcal{T}$. Entities in sentences are labeled with types drawn from type inventory $\mathcal{T}$.

Type set $\mathcal{T}$ is consist of hierarchical structure types. For example, type hierarchies take the form of a forest, where each tree is rooted by a top-level supertype (e.g.,/person, /person/artist). The assignment of labels can be represented as: $y_k, y_K = 1$ if $k, K \in \mathcal{T}$ and $k \in K$.

We now introduce notations for referring to aspects of hierarchical types in this paper. Relation "type k is a subtype of K" is denoted as $k \in K$. Relation "m is one of the siblings of k, which shares the same root" is denoted $y \in Sb(y)$ where $Sb(y) \subseteq Y$.

### 3.2 OVERVIEW OF MODEL

Our model represents mentions and entity types as hRME. (Figure 1) Given sentence text $\mathcal{S}$ with marked mention span $\mathcal{X}$ and types $\mathcal{T}$ respectively, first we employ language model Bert or RoBERTa (Liu et al., 2019) to encode input sentence with mention span into a single vector. Second, to generate the hyper-rectangle mass representation from the vector, hyper-rectangle projection and mass representation modules transform representations in vector space to the hRM space which consists of four parameters $(x_d, x_s, x_i, x_e)$ (These parameters mean density, size scale value, initial point, and endpoint on an axis respectively). Once the single vector is transformed into hRM representation, the hierarchical classifier step computes the conditional probability of the type $t \in \mathcal{T}$ given the context and mention $(\mathcal{S}, \mathcal{X})$. The last step is the adaptive threshold, which allocates different threshold values to each type and thus logits will be compared with the threshold to determine the suitable types for each input sentence and mention.

### 3.3 HRM REPRESENTATION

In total, hRM is characterized by four parameters $(x_d, x_s, x_i, x_e)$, where $x_d \in \mathbb{R}$ is the density of hRM, $x_s, x_i, x_e \in \mathbb{R}^n$ are the size scale, initial corner and end corner of the hyper-rectangle (Figure 3).

**Density parameter** The density parameter $x_d$ is the density for each hRM. It is restricted that types share same root have same density.

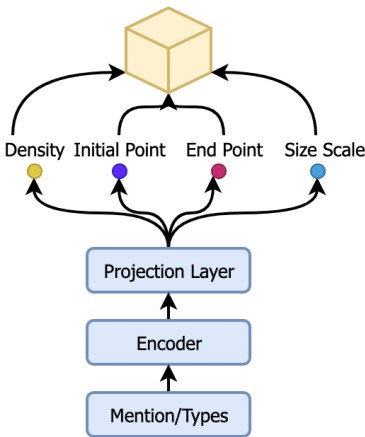

Figure 3: A natural language is embedded into hRM

**Size scale parameter** The size scale parameter $x_s$ is used to multiply by the edge. It is assumed that edge length is linear to the type level. For example, root type deserves a larger size than other level types.

**Initial and end corner** The initial corner and end corner determined the size of the rectangle in 2D view. Figure4 shows the illustration of size scale, initial and end corner.

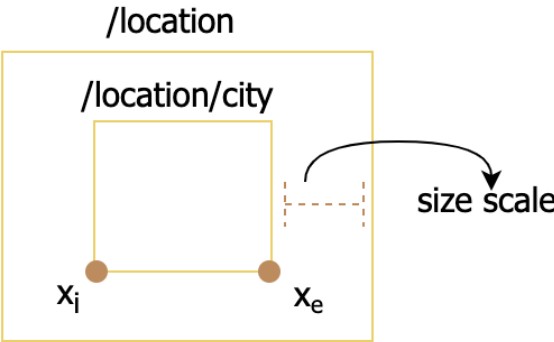

Figure 4: 2D view of hRM with size scale, initial corner and end corner

**Projection layer** Projection layer extracts features from the hidden vector. We use highway network (Srivastava et al., 2015) which projects single vector into hRM. Besides, we use the softplus function (Zheng et al., 2015) to guarantee hRM is positive. To ensure that gradients flow for all conditions, we use Gumbel distributions (Dasgupta et al., 2020) rather than Gaussian convolution (Li et al., 2019). Gaussian-box process (Rasmussen & Williams, 2005) is the most natural way to represent hRM. However, it cannot support us to solve the expected hRM in an elegant form. We choose gumbel-box process instead to mitigate learning difficulties arising from model unidentifiability and the attendant large flat regions of parameter space. Following the gaussian-box process, we use softplus approximation to represent the mass of hyper-rectangle and train embedding later.

**hRM representation** In total, hRM is then computed as:

$$\mathbf{Ma}(x) = x_d \prod_k ((x_{i,k} - x_{e,k})x_s) \tag{1}$$

### 3.3.1 MENTION REPRESENTATION

To represent a sentence with a mention in mass representation, we first chunk into tokens with a specific format. Given a mention span $m$ in a sequence of words denoted as $s$, We format input with tokens [CLS] and[SEP] as :

$$\text{input} = [\text{CLS}]\text{s}_1[\text{SEP}]\text{m}[\text{SEP}]\text{s}_2 \tag{2}$$

Further, the encoder (BERT-large model) (Devlin et al., 2018) encodes chunked tokens into feature vectors. We encode the whole sequence into a single vector and [CLS] token is chosen to take as the hidden vector.

Moreover, we split the hidden vector and then project them into mass representations. Similarly, softplus is used to guarantee the mass of each input is positive.

We approximate the mention as the following equation with softplus function:

$$\mathbf{Ma}(x) = \prod \text{softplus}(\frac{(x_e - x_i) * x_s}{\beta} - 2\gamma) * x_d \tag{3}$$

### 3.3.2 HIERARCHICAL TYPE REPRESENTATION

In different, density and size scale parameters are restricted compared with section3.3.2. Types that share the same root hold the same density but different size scales (Figure 5). See figure6 for more details about edge length.

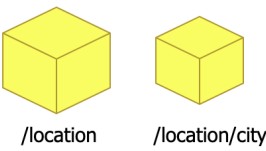

Figure 5: `/location` and `/location/city` share same density(same color), but different size scale

### 3.4 HIERARCHICAL LEARNING LOSS

We aim to minimize the combination of losses for various hierarchies with the proposed model structure.

In previous works, a linear combination of losses is normally employed and constant weight is assigned to each loss function (Beltagy et al., 2020). However, it has two drawbacks: (1) It is only possible to achieve an optimal solution when the Pareto frontier is convex (Boyd & Vandenberghe, 2004)). (2) Weight assigned to each loss function may not reflect the actual behavior.

We develop another loss function: hypervolume indicator (Miranda & Von Zuben, 2015). This metric preserves the optimal point in various conditions including non-convex Pareto frontiers. To get rid of Nadir point (Miettinen et al., 2010) in original equation of hyper-volume indicator, we simplify the logarithm equation as:

$$logH_z(x) = \sum_{i=1}^{n} log(0.5 - f_i(x)) \tag{4}$$

Compared with prior work, the Nadir point is simplified to 0.5 and thus computation becomes easier. In addition, it achieves a better result than the linear combination of losses.

## 3.5 ADAPTIVE THRESHOLDS

In prior works, a fixed threshold ($r = 0.5$) is used for classification of all types (Onoe et al., 2021). For severe type imbalance circumstances, all types share a fixed threshold that leads to poor performance. To find threshold values for each type, a previous approach employs a threshold searching system that searches in 20 evenly spaced numbers over a specified interval [0,1] in each evaluation period. The best threshold is then determined from the best F1 on the dev set (Zhang et al., 2018). However, it has a few drawbacks:

(1) the threshold searching system is time-consuming;

(2) the true threshold is not in evenly spaced numbers.

To overcome drawbacks in prior works, we instead find a time-saving and more precise threshold searching system. In the initial iteration, the threshold for each type is evaluated in 10 evenly spaced numbers within [0,1] and the best threshold maximizes the strict F1 on the dev set. Once the best threshold($r_b$) is found in initial evaluation on the dev set, we then assume the best threshold as center and threshold interval to [$r_b$-0.25,$r_b$+0.25]. Then in the following evaluations, the best threshold is selected from 10 randomly sampled points from the prior interval [$r_b$-0.25,$r_b$+0.25] which maximizes the strict F1 on the dev set.

## 4 EXPERIMENTS

### 4.1 BASELINE

Our chief comparison is box embedding modeling of entity types (Onoe et al., 2021). As the primary baseline for all experiments, we use the same **box-based** version in both box embedding and our model.

### 4.2 DATASETS

We conduct experiments on three standard fine-grained entity typing datasets: Figer, UFET, and OntoNotes.

#### 4.2.1 FIGER

Figer dataset (Ling & Weld, 2021) is a fine-grained set of two hierarchical levels and 112 types, formulating the tagging problem as a multi-class and multi-label classification task. Sentences are sampled from Wikipedia articles and news reports.

#### 4.2.2 UFET

Ultra-Fine Entity typing dataset (Choi et al., 2018) has three types: general, fine-grained, and ultra-fine-grained types. UFET dataset is crowd-sourced and on average, each example in UFET has 5 labels: 0.9 general, 0.6 fine-grained, and 3.9 ultra-fine types. It is noticeable that classification in UFET does not provide explicit hierarchies in types, and thus all classes are treated equally during training and evaluation.

#### 4.2.3 ONTONOTES

OntoNotes (Weischedel et al., 2011) is a large multilingual richly-annotated corpus constructed at 90% inter-annotator agreement. For entity typing task, it has three hierarchical levels and 89 types in total.

### 4.3 EVALUATION METRICS

We follow the standard evaluation protocol and use normal evaluation metrics. For Figer and OntoNotes, we adopt the metrics used in (Ling & Weld, 2021) where results are evaluated via loose macro and loose micro F1 scores. For UFET dataset, we adopt macro-precision, macro-recall, and macro-F1.

## 4.4 IMPLEMENTATIONS

We use bert-base uncased and bert-large uncased as two base encoders for tasks, for a fair comparison with previous work and an investigation of baseline models. We also use the same parameters compared with the baseline box embedding model. For example: All experiments project onto a target space of 109 dimensions, softplus scale=1, type box = CenterSigmoidBoxTensor. Apart from similar parameters, we set typical parameters: size scale=0.5, threshold step = 0.05.

## 5 RESULTS AND ANALYSIS

We compare the experimental results of our approach with previous models. In addition, we study the contributions of our base model architecture, hypervolume indicator, and adaptive thresholds via ablation. To ensure our findings are reliable, we run each experiment multiple times and report the average performance.

Overall, our approach significantly increases the state-of-the-art F1 on Figer and parts of UFET datasets. Besides, compared with the chief comparison model, our approach improves on all three datasets.

### 5.1 FIGER

| Model | Ma-F1 | Mi-F1 |
|---|---|---|
| Box Embedding(Onoe et al., 2021) | 79.4 | 75.0 |
| HET(Chen et al., 2020) | 82.6 | 80.8 |
| FGET-LTR(Lin & Ji, 2019) | 83.0 | 79.8 |
| LITE(Li et al., 2022) | 80.1 | 83.3 |
| Our Model | **84.9** | **83.5** |
| w/o Adaptive thresholds | 83.8 | 80.9 |
| w/o Hyper-volume loss function | 80.2 | 78.5 |

Table 1: Performance on the Figer dataset

Table 1 shows the macro-F1 and micro-F1 scores on the Figer test set. Our hRMM outperforms the baseline box model (Onoe et al., 2021) on both Ma-F1 and Mi-F1 and holds the best Ma-F1 and Mi-F1 in state-of-the-art systems. Our model can clearly distinguish the root type and subset type(see example in Appendix A) We notice that removing adaptive thresholds only causes a small performance drop. It holds a similar condition with OntoNotes which might be because of the systematic shifts between training and test distribution.

### 5.2 UFET

| Model | Total | | | Coarse | | | Fine | | | Ultra-Fine | | |
|---|---|---|---|---|---|---|---|---|---|---|---|---|
| | P | R | F1 | P | R | F1 | P | R | F1 | P | R | F1 |
| Box(Onoe et al., 2021) | 52.9 | **39.1** | 45.0 | 71.2 | **82.5** | 76.4 | 50.9 | 55.2 | 53.0 | 45.4 | 24.5 | 31.9 |
| LDET(Onoe & Durrett, 2019) | 51.6 | 32.8 | 40.1 | 67.4 | 80.6 | 73.4 | 41.6 | 54.7 | 47.3 | 46.3 | 15.6 | 23.4 |
| LRN(Liu et al., 2021) | 53.7 | 38.6 | 44.9 | **77.8** | 76.4 | **77.1** | **55.8** | 50.6 | 53.0 | 43.4 | **26.0** | 32.5 |
| HY Xlarge(López & Strube, 2020) | 43.4 | 34.2 | 38.2 | 61.4 | 73.9 | 67.1 | 35.7 | 46.6 | 40.4 | 36.5 | 19.9 | 25.7 |
| Our Model | **53.9** | 38.7 | **45.5** | 71.2 | 82.1 | 76.3 | 51.5 | **55.4** | **53.4** | **46.9** | 24.9 | **32.6** |
| w/o Adaptive thresholds | 53.3 | 38.9 | 45.0 | 71.0 | 81.9 | 76.1 | 51.1 | 55.1 | 53.0 | 45.9 | 24.7 | 32.1 |

Table 2: Performance on the UFET dataset

Table 2 shows the main results of all baselines and our method on the UFET test set. For a fair comparison, we implement the baseline and our approach with same encoder in all experiments. We can see that:

1) hRM representation still achieves the best performance on non-hierarchical knowledge scenarios which verifies the calibration of our method.

2) Both density and size scale are two useful parameters for fine-grained label prediction. Compared with simple box embedding, hRMM can achieve 0.5% F1 improvement by adding these two parameters to the Ultra-Fine type.

Lack of prior hierarchical knowledge leads to no hypervolume indicator, We believe it will leverage label dependencies better to predict labels once hierarchical knowledge exists.

## 5.3 ONTONOTES

| Model | Ma-F1 | Mi-F1 |
|---|---|---|
| Box Embedding(Dasgupta et al., 2020) | 77.3 | 70.9 |
| MLM-ET(Dai et al., 2021) | **85.44** | **80.35** |
| HYM-ET(López & Strube, 2020) | 82.0 | 80.2 |
| LR-ET(Liu et al., 2021) | 77.6 | 71.8 |
| LABELGCN(Xiong et al., 2019) | 77.8 | 72.2 |
| PLE(Ren et al., 2016) | 79.8 | 77.8 |
| Our Model | 83.2 | 76.5 |
|     w/o Adaptive thresholds | 82.6 | 75.9 |
|     w/o Hyper-volume loss function | 80.9 | 73.1 |

Table 3: Performance on the OntoNotes dataset

We further conduct experiments on OntoNotes and report results without augmentation data in Table 3.

We consider that: (1) hRMM achieves better performance on both metrics compared with the baseline simple box model (Onoe et al., 2021). (2) Our method is not leading the board compared with state-of-the-art models. We found the main reason is: 52% of mentions were marked as \Other in the dataset, which causes difficulty for our model to generate hyper-rectangles that distinguish from other types.

In addition, long-tailed problems and less amount of types are more severe in OntoNotes compared with Figer and UFET cause hRMM cannot learn types well.

## 5.4 DISCUSSION OF HIERARCHIES

In our model, hierarchy learning is improved by size scale and density parameters compared with baseline model. Size scale restricts the hRM size and thus keeps subset type inside its root type. For density parameter, same root types share the same density value. It guarantees that the mass of the root type is larger than its subtypes. As result, the following figure6 shows edges for root type /person and its subtype /person/author. In conclusion, restricted hierarchies have a significant superior rather than prior investigations.

## 5.5 ERROR ANALYSIS

### 5.5.1 DATA INCONSISTENT

Data error influences the model metric. In the OntoNotes dataset, we notice that some of the examples have inconsistent labels because of the distant supervision of data labeling. Here we show a few data errors.

(1) /organization/sports team is present, but its root type /organization is missing, which penalizes models that predict the root type correctly

(2) Present of /other with /organization/company and /organization, which confuses model while training.

Besides, in OntoNotes, 52% of mention spans were marked as /other. There is not a clear definition of /other type leading confusions to model.

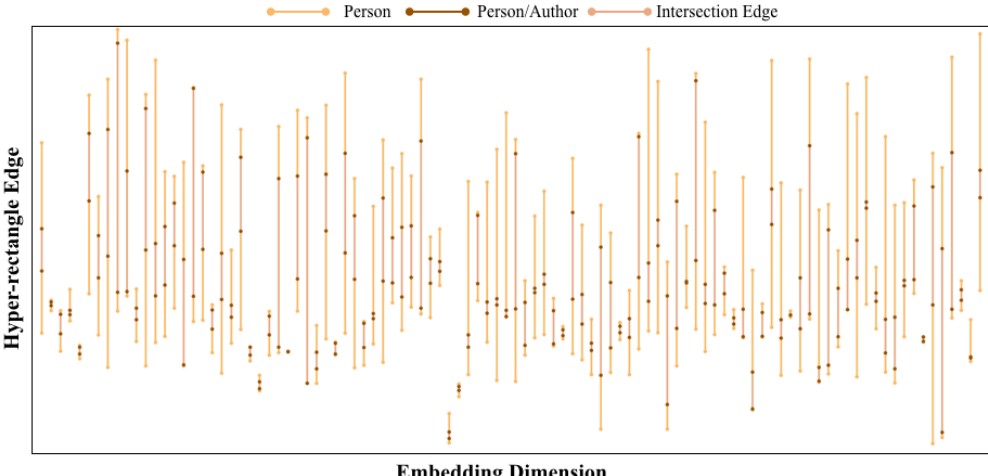

Figure 6: Edges of `/Person` VS `/Person/Author`. Because `/Person` is the root type of `/Person/Author`, edges of subtype are all covered by root type

### 5.5.2 DATA IMBALANCE

Data imbalance is an another reason for leading to model prediction error. For Figer dataset, `/location`, `/person` and `/organization` are three most types. Besides, it is calculated that the top 3 types cover over 80% of the evaluation data.

## 6 CONCLUSION

In this paper, we propose a new approach for fine-grained entity typing. There are three contributions:

(1) We propose a brand new architecture that represents natural language in hRME.

(2) We adopt hypervolume indicator rather than the linear combination of losses to achieve a better loss function.

(3) We utilize adaptive classification thresholds to further boost the performance.

Experiments show the benefits of our model over prior investigations and we achieve new state-of-the-art results on two benchmarks. For future work, we are interested in incorporating additional ideas from contrast learning and federated learning. Furthermore, we are also passioned about adapting our model to other NLP domains

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

## A  APPENDIX

We plot mention `Apple Company`, type `/Organization` and unrelated type `/Organization/Educational Institute` respectively in Figure 7. Compared with two figures, Intersection of Figure 7(a) is largely more than Figure 7(b)

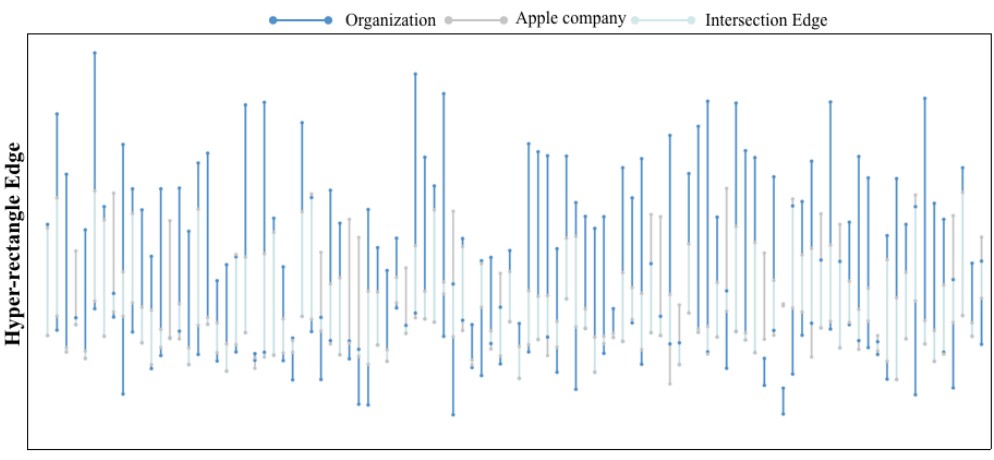

(a) Edges of `/Organization` VS `Apple company`

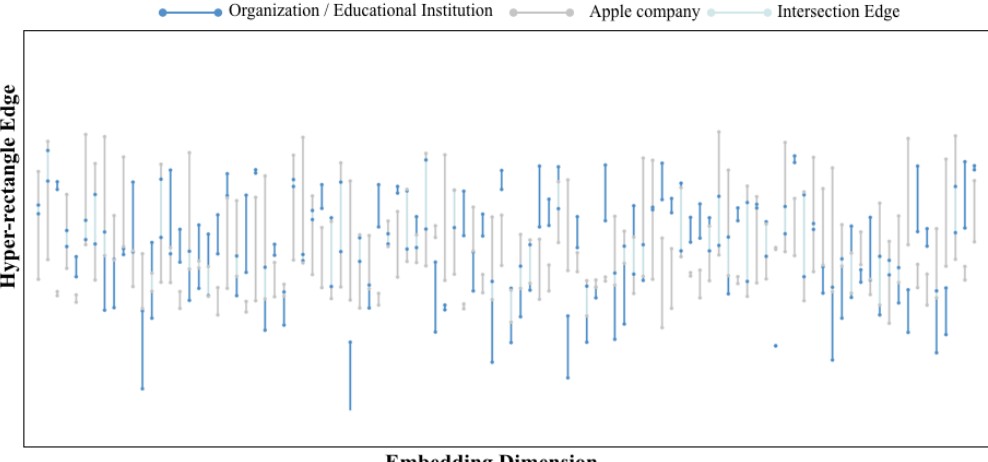

(b) Edges of `/Organization/Educational Institute` VS `Apple company`

Figure 7: Comparison of same mention (grey line) between different hierarchical types (blue lines)

