# OpenReview forum: "A Hierarchical Hyper-rectangle Mass Model for Fine-grained Entity Typing"
_ICLR.cc/2023/Conference — Submitted to ICLR 2023_

### Official Review · Reviewer_6W9o · 2022-10-25

**Confidence:** 3
**Correctness:** 2
**Technical Novelty And Significance:** 2
**Empirical Novelty And Significance:** 2
**Recommendation:** 3

**Clarity, Quality, Novelty And Reproducibility:**

I was unclear on how the methods worked, how the hierarchy was encoded, or why we would expect this approach to outperform box embeddings.

I was unclear about how the hierarchy is actually enforced in the model.  Section 3.4 is titled “hierarchical learning loss” but the ‘hypervolume indicator’ discussed there is never actually defined.  I never understood how the hypercubes defined in equations 1 and 3 were actually used in the model, or whether/how the model might constrain the boxes for hierarchically-organized labels to nest under their parent label boxes.  It could be that the paper is assuming that the reader has knowledge about how the baseline box embeddings are typically used in NER models, which I lacked.

Further adding to my confusion, after the experiments, section 5.4 Discussion of Hierarchies talks about constraints on the type space that might exist in the model.  For example, it says “Size scale restricts the hRM size and thus keeps subset type inside its root type.”  I was confused, is this an empirical fact that is observed in the trained model, or is this built in to the model?  Further, Figure 6 seems to be illustrating how the hierarchy is reflected in the model, but is very hard to read and understand.  What is an “intersection edge”?  I expected to see here that the person/author box is contained within the person box, but I don’t see the plot including any color as dark as the person/author color in the legend, but it could be just that the line width is lower in the plot than in the legend. [edit: after further reflection I think I understand Fig 6, but it could be explained better]

I think the paper could be greatly improved by presenting box embeddings in technical detail, and then explicitly contrasting that method with the paper’s proposed one (talking about how the size and density parameters are added, and why these are critical).  Then, the experimental analysis should show that the hypothesized source of improvement over box embeddings is in fact leading to the experimental wins.  The paper says in the intro that box embeddings are too simple and can’t learn hierarchical knowledge, but I found these claims to be unsubstantiated.


**Strength And Weaknesses:**

Strengths

The results in this paper are strong, over a number of data sets, and better handling of hierarchically-organized types in NER is an important task.

Weaknesses

I found the paper to be so unclear that I did not have confidence in the contribution (as discussed below), and to the extent I understood the method I found the novelty to be limited.  The model appears to use box embeddings but with additional size and density parameters which are a fairly straightforward addition.  If the motivation for including these had been described more clearly, it's possible even with limited novelty this could be a sufficient contribution for publication.

**Summary Of The Paper:**

This paper presents an entity typing technique that uses geometric embeddings, and evaluates it on several data sets, achieving strong performance.

**Summary Of The Review:**

While this paper goes after an interesting task, I found the clarity and novelty to be too limited for publication.

---

### Official Review · Reviewer_Q73W · 2022-10-25

**Confidence:** 4
**Clarity, Quality, Novelty And Reproducibility:** refer to the weakness part
**Correctness:** 3
**Technical Novelty And Significance:** 2
**Empirical Novelty And Significance:** 2
**Recommendation:** 5

**Strength And Weaknesses:**


Strength

- The additional parameter: density and edge scaler make sense for a mass model to represent the hierarchical information in types.
- The method works well on FIGER

Weakness

- Baselines such as LITE (included in the FIGER datasets, SOTA, 50.6 ma-f1), and MLM-ET (49+ ma-f1) are not included in the UFET benchmarks, authors should clarify why you discard them before claiming that **you are SOTA**.
- The total model is too close to the Box-embedding for FET (Modeling fine-grained entity types with box embeddings. Onoe et al,. 2021), basically adding a scaler and a vector multiplier for type and mentioned encoding, the contribution is not big. The performance of hRMM is subtle on UFET (0.5 ma-f1 higher than Box embedding with “adaptive threshold”), especially given that the SOTA model is LITE with 50.6 ma-f1.
- The adaptive threshold part is a simple trick that doesn’t have too much contribution.
- The density scaler and the size scaler vector is the key difference between the basic box embedding (Onoe et al,. 2021),  however, the authors lack a detailed introduction and analysis of them. Why the density x_d \in \mathcal{R}? It should be positive x_d \in \mathcal{R^+}?  Is the size scaler the same for all edges in a hyper-rectangle? It should be positive? x_d and x_s are both trainable parameters or fixed during training?

Minors:
- parenthesis usage: hierarchical hyper-rectangle mass model(hRMM) → hierarchical hyper-rectangle mass model (hRMM), should leave a space.
- Figure2 → Figure 2, Figure4 → Figure 4

**Summary Of The Paper:**

This paper proposes hRMM for fine-grained entity typing, which represents mentions and types into the hyper-rectangle mass to capture the relationships of ontology into a geometric mass view.

**Summary Of The Review:**

This paper proposes a new model for fine-grained entity typing, which represents mentions and types into the hyper-rectangle mass to capture the relationships of ontology into a geometric mass view. The novelty seems limited and the empirical results are relative weak.

---

### Official Review · Reviewer_W1hi · 2022-10-25

**Confidence:** 4
**Correctness:** 2
**Technical Novelty And Significance:** 2
**Empirical Novelty And Significance:** 2
**Recommendation:** 3

**Clarity, Quality, Novelty And Reproducibility:**

The paper is not clear and written in poor quality. The novelty is limited. The details of the experiments are not clear which makes it difficult to reproduce the main results.

**Strength And Weaknesses:**

Strength:

The authors introduced density and size scale parameters to improve the representation ability of box embedding.

Weaknesses

1. Weak motivation and novelty: while the authors propose to introduce density and size scale to box embedding, there is no clear explanation why we need these two parameters. The original box embedding has already been able to capture the inter-dependencies among event types.

2. The experiments are not solid. In terms of performance gain, the proposed approach only achieves marginal improvements over the baselines on two of the datasets, and underperforms the previous baseline on OntoNotes. Also, there is no detailed empirical analysis of the effect of the two new parameters.

3. The reported results of LITE in Table 1 are not consistent with the results reported in the original paper. In fact, the results in Table 1 are merged from the two versions of LITE (pre-trained on NLI+UFET + NLI+task-specific training). It seems this is a technical error.

4. Writing needs to be significantly improved. The paper is poorly structured and lacks many details, e.g., the intuition and motivation of the approach, as well as theoretical or empirical analysis.


**Summary Of The Paper:**

This paper proposes to introduce two additional parameters – density and size scale in the box embedding for fine-grained entity typing. The experiments on three benchmarks show the proposed approach achieved some marginal improvements over the considered baselines.

**Summary Of The Review:**

This paper has poor quality with inadequate empirical or theoretical support. The reported results on some of the baselines may have technical flaws. The writing is poor and unclear. The results may not be easily reproduced.

---

### Official Review · Reviewer_zfGa · 2022-10-28

**Confidence:** 4
**Correctness:** 2
**Technical Novelty And Significance:** 3
**Empirical Novelty And Significance:** 3
**Recommendation:** 3

**Clarity, Quality, Novelty And Reproducibility:**

#### Clarity & Reproducibility
I found this paper quite hard to follow, despite having some familiarity with the tasks and related methods.

A lot of core concepts are only described vaguely in text and the losses used during training are never properly defined. I think that the mass representations detailed in equations (1) and (3) are probably applied to the intersection of mention and type representations (illustrated in Figure 2), but this is never explicitly stated. Instead the mass representation is described as a mention representation only. The hyper-volume loss function seems to be essential but, again, I don't understand the definition in Equation (4). What is $f_i(x)$ here? Section 3.3.2 does not properly describe how the size scale and density parameters are used to enforce the type hierarchy (and it seems to refer back to itself?).

#### Quality & Novelty
I feel that there might be something interesting and novel here but, given my concerns above, it is hard to properly say.

**Strength And Weaknesses:**

#### Strengths
- The results are quite positive with respect to recent published work.
- The claim of building hierarchical representations without complex training procedures, is an interesting one.

#### Weaknesses
- The current presentation lacks a lot of essential detail and motivation. Neither of the key concepts 'size scale' and 'density' are properly motivated and the full model & losses are not mathematically defined. I have more comments below in the section on clarity & reproducability.


**Summary Of The Paper:**

This paper presents a new method of hierarchical entity typing that represents mentions and types using a hyper-rectangle representation, models the mention-type relation using the intersection of these representations, and classifies per-mention types according to the mass of this intersection. The mass representation differs from related work in box embeddings, with the introduction of size scale and density paramaters. There is also a novel loss formulation to aid learning of hierarchical representations.

The new approach outperforms previous work on two fine grained entity typing benchmarks (Figer, UFET) and approaches state of the art on OntoNotes. A visualization of the hyper-rectangle representation of the types `/person` and `/person/author` show that the more general type subsumes the more specific one, geometrically.


**Summary Of The Review:**

This paper presents a method of hierachical entity typing that achieves impressive results, with respect to recent published work. However, the current presentation is unclear and key concepts are not properly motivated or defined. I feel that Section 3 needs to be rewritten to (a) properly motivate the size scale and density parameters, (b) mathematically define the full process of hierarchical entity type classification, (c) fully define the loss used to train the model.

---

### Decision · Program_Chairs · 2023-01-20

**Decision:**

Reject

**Justification For Why Not Higher Score:**

There are concerns regarding motivation of the approach, as well as significance of results.

**Justification For Why Not Lower Score:**

n/a

**Metareview: Summary, Strengths And Weaknesses:**

The paper studies the problem of fine-grained entity typing.
The paper proposes having two additional parameters,  density and size scale in the previously introducing box embeddings.
The reviewers have concerns regarding: 1) lack of motivation behind the proposed solution 2) marginal improvements in results 3) lack of  presentation clarity.   The authors are encouraged to continue their work, address reviewer concerns, and  resubmit at a future conference at ICLR or other venues.

**Summary Of Ac-Reviewer Meeting:**

n/a